# Ohmic transition at contacts key to maximizing fill factor and performance of organic solar cells

Jun-Kai Tan[1,2], Rui-Qi Png[1,2], Chao Zhao[1,2] & Peter K.H. Ho[1,2]

While thermodynamic detailed balance limits the maximum power conversion efficiency of a solar cell, the quality of its contacts can further limit the actual efficiency. The criteria for good contacts to organic semiconductors, however, are not well understood. Here, by tuning the work function of poly(3,4-ethylenedioxythiophene) hole collection layers in fine steps across the Fermi-level pinning threshold of the model photoactive layer, poly(3-hexylthiophene):phenyl-$C_{61}$-butyrate methyl ester, in organic solar cells, we obtain direct evidence for a non-ohmic to ohmic transition at the hole contact that lies 0.3 eV beyond its Fermi-level pinning transition. This second transition corresponds to reduction of the photocurrent extraction resistance below the bulk resistance of the cell. Current detailed balance analysis reveals that this extraction resistance is the counterpart of injection resistance, and the measured characteristics are manifestations of charge carrier hopping across the interface. Achieving ohmic transition at both contacts is key to maximizing fill factor without compromising open-circuit voltage nor short-circuit current of the solar cell.

[1] Department of Physics, National University of Singapore, Lower Kent Ridge Road, Singapore S117550, Singapore. [2] Solar Energy Research Institute of Singapore, National University of Singapore, Engineering Drive 1, Singapore S117574, Singapore. Correspondence and requests for materials should be addressed to P.K.H.H. (email: phyhop@nus.edu.sg)

The power conversion efficiencies (PCE) of organic solar cells have recently breached the 10% benchmark in both small-molecule- and polymer-based organic solar cells with single-junction photoactive layers (PAL)[1]. This stimulates renewed interest in urban solar, indoor, and wearable applications and also in tandem organic–silicon hybrid cell structures. As the short-circuit current density ($J_{sc}$) of an organic solar cell reaches the photogeneration limit of its donor–acceptor PAL, further improvement in performance can only come from increasing its open-circuit voltage ($V_{oc}$) and fill factor (FF). Therefore numerous studies have focused on the underlying physics of $V_{oc}$[2–6] and FF[7–9]. In particular, detailed analyses of the cell voltage deficit referenced to the semiconductor charge-transfer gap have been attempted, i.e., $\Delta = E_{CT}/e - V_{oc}$, where $e$ is the electronic charge, and $E_{CT}$ is the energy difference between the band edges of the donor highest-occupied molecular orbital and the acceptor lowest-unoccupied molecular orbital (LUMO)[2–6].

In addition, contacts can further limit what is thermodynamically possible in the bulk of the cell. The role of contacts between the PAL and each of the two electrodes has not been well understood. Often cell properties are analyzed assuming that contacts do not limit behavior. However, contacts set the diode built-in potential $V_{bi}$, which does limit the attainable $V_{oc}$[10–12]. Clearly $V_{oc}$ cannot significantly exceed the illuminated $V_{bi}$, as the internal field will reverse to oppose power generation. The notion that $V_{bi}$ limits $V_{oc}$ is not surprising, for $V_{bi}$ determines the injection current–voltage characteristic of the cell. Two constituent voltage deficits may thus be defined: $\Delta_1 = V_{bi} - V_{oc}$, which turns out to be small, of the order of a few $kT/e$[10]; and $\Delta_2 = E_{CT}/e - V_{bi}$, which turns out to be large, of the order of a few tenths of a volt, due to the combined effects of Fermi-level (FL) pinning, polarization band-bending, and electrostatic band-bending losses at the contacts[10], suggesting much room for improvement through careful design of these contacts[11].

Yet only rudimentary understanding is available for the desired ohmic contact. By definition, such a contact exhibits a contact resistivity that is small compared to the bulk resistivity of the device. Little is known however about contact resistivity of organic semiconductors, and their ohmic contact criteria beyond the requirement for shallow FL pinning to the relevant band edge of the semiconductor[13,14]. However, FL pinning relates to charge transfer equilibrium[15–17], while ohmic contact refers to charge transfer kinetics; the two phenomena are not synonymous[18]. Detailed investigations of these questions have been hampered by the lack of suitable model electrodes whose work function $\phi$ can be varied in fine steps over a wide range, without accompanying changes in their morphology or doping density that would complicate analysis.

Here we have developed a family of hole-doped poly(3,4-ethylenedioxythiophene)-based polymers (PEDT:PSSCs$_x$H$_{1-x}$) to provide the requisite model electrodes. We used these as the hole collection layers (HCL) to investigate both hole injection and collection in organic solar cells with poly(3-hexylthiophene):phenyl-C$_{61}$-butyrate methyl ester (P3HT:PCBM) as the PAL. We show that besides setting $V_{bi}$, the work function of the charge collection layer can also limit FF through the charge-extraction resistance at the contact. We chose P3HT:PCBM because of its good batch-to-batch repeatability and high carrier mobilities that yield non-transport-limited photocurrents[8]. In fact, P3HT:PCBM is a high quantum-efficiency PAL, with internal photon-to-electron conversion efficiency approaching 0.9[19]. By tuning the HCL $\phi$ in fine steps over the range $4.4 \lesssim \phi \lesssim 5.2$ eV, we first established the work function for the onset of FL pinning $\phi_{pin}$ and then showed that while both $V_{bi}$ and $V_{oc}$ level off beyond $\phi_{pin}$, FF continues to improve until a second threshold is reached, which we denote the optimal work function $\phi_{op}$. The product of $V_{oc}$ and

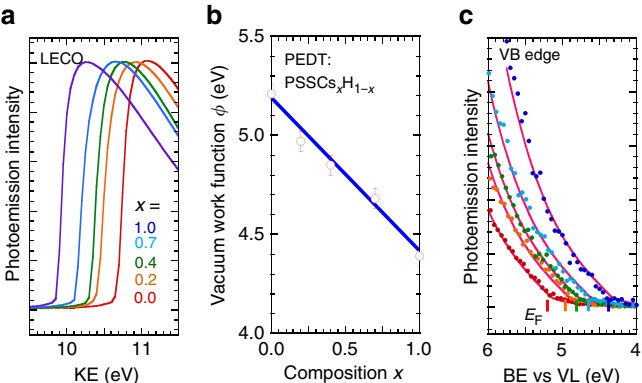

**Fig. 1** UPS of PEDT:PSSCs$_x$H$_{1-x}$ films. **a** Normalized ultraviolet photoemission spectra, showing the low-energy cutoff (LECO) region plotted against kinetic energy (KE) of the emitted photoelectrons ($h\nu$, 21.21 eV) for films with different $x$. **b** Plot of vacuum work function $\phi$ against composition $x$. Blue line gives the linear fit, $\phi = 5.2 - 0.8x$. Error bar gives the standard error. **c** Valence band (VB) edge region, plotted against binding energy (BE) relative to vacuum level (VL). Fermi energy ($E_F$) is marked for each film. Dots are data (color coding same as in **a**); magenta lines are global fits to a master shape function as guide to the eye

FF, and hence PCE, maximizes at this $\phi_{op}$, which lies ca. 0.3 eV beyond $\phi_{pin}$. We then developed a general methodology to extract contact resistivity $\rho_c$. We showed from current detailed balance theory that this $\rho_c$ corresponds to the intrinsic charge-transfer resistance at the semiconductor/electrode interface. Finally, we established that $\phi_{op}$ marks the ohmic transition that arises from the strong exponential decrease of $\rho_c$ with work function. While the deleterious effects of series resistance, whether contact or otherwise, are well known[10,12,20], the existence of this intrinsic source of contact resistance has not been appreciated, for it cannot be elucidated from fitting to phenomenological expressions[9,21,22].

## Results

**Tunable work function films**. We have developed the hole-doped PEDT:PSSCs$_x$H$_{1-x}$ system to give composition-tunable $\phi$ over a 0.8-eV-wide range by extending our previous work on spectator-ion effects[23]. Figure 1a shows the low-energy cutoff (LECO) region of selected PEDT:PSSCs$_x$H$_{1-x}$ films measured by ultraviolet photoemission spectroscopy (UPS). $\phi$ was obtained in the usual way from the photoelectron kinetic energy difference between LECO and FL: $\phi = KE_{LECO} - KE_{FL} + h\nu$, where $h\nu$ is the He I photon energy (21.21 eV). As $x$ increases from zero to unity, $\phi$ decreases from 5.2 eV to 4.4 eV; linear regression gives $\phi = 5.2 - 0.8x$ (Fig. 1b). This $\phi$ shift arises from the local Madelung potential effect of the mixed $^-SO_3H/^-SO_3^-Cs^+$ local ion cluster on the electrochemical potential of the hole carriers[23,24]. The valence band edge of PEDT shifts rigidly with FL, which excludes any change in its electronic structure or doping level (Fig. 1c). This ability to vary work function without altering carrier density crucially avoids changing resistivity, Schottky barrier width and band bending in the HCL. The valence band features of PEDT, PSS, and Cs$^+$ are also fixed in energy relative to the vacuum level, which excludes any change in the surface-dipole component of the work function (Supplementary Figures 1a and b). $JV$ measurements on indium tin oxide (ITO)/45-nm PEDT:PSSCs$_x$H$_{1-x}$/Al sandwich structures gave resistivity of <0.3 $\Omega$ cm$^2$. The PEDT:PSSCs$_x$H$_{1-x}$ system thus provides model electrodes with well-defined work functions that can be manipulated freely without altering surface morphology, surface dipole, or doping density of the films.

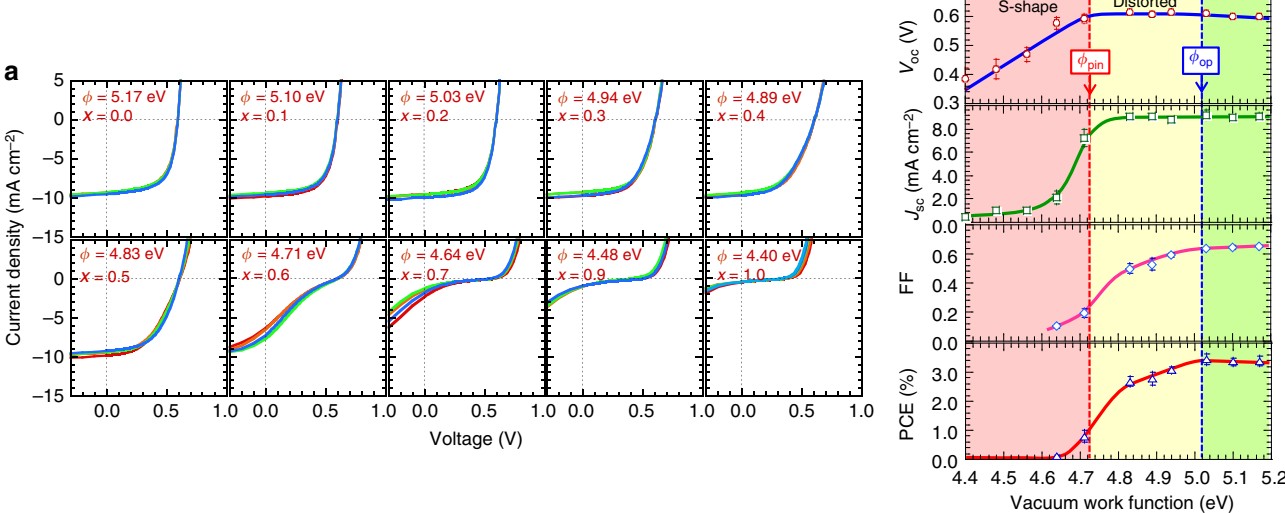

**Fig. 2** Characteristics of organic solar cells with different hole collector work functions. **a** JV characteristics of ITO/PEDT:PSSCs$_x$H$_{1-x}$/P3HT:PCBM (1.0:0.8 w/w)/Ca/Al cells measured at 110-mW cm$^{-2}$ illumination, spectral-mismatch corrected. Four diodes are shown in each panel. Vacuum work function $\phi$ and composition $x$ of the PEDT:PSSCs$_x$H$_{1-x}$ hole collection layers are given. **b** Cell performance parameters plotted against vacuum work function of the hole collection layer. Symbols are data; lines are guides to the eye. Error bar gives the standard error. The standard error in work function is 0.03 eV. The Fermi-level pinning work function $\phi_{pin}$ and optimal work function $\phi_{op}$ are indicated. "S-shape" and "Distorted" describe the shape of the illuminated JV characteristics in **a**

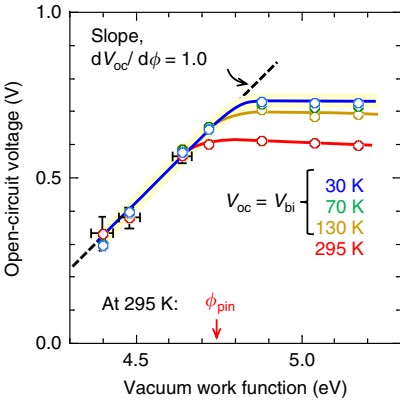

**Fig. 3** Built-in potential of organic solar cells with different hole collector work functions. Cell open-circuit voltage plotted against vacuum work function of the hole collection layer. Conditions: 1.1-sun illumination; temperature, 295 K → 30 K. For $T \lesssim 150$ K, $V_{oc}$ corresponds to $V_{bi}$; for $T \lesssim 30$ K, $V_{bi}$ approaches the zero-Kelvin value $V_o$ (0.73 V). Typical uncertainties: for large $\phi$: $V_{oc}$, ±0.01 V; for small $\phi$: $V_{oc}$, ±0.05 V. $V_{oc}$ data are averaged for forward and reverse sweeps

**Solar cells with different hole collector work functions**. P3HT: PCBM diodes were then fabricated by spinning 85-nm-thick P3HT:PCBM films, with composition 1.0: 0.8 weight/weight (w/ w), over 45-nm-thick PEDT:PSSCs$_x$H$_{1-x}$ films that were coated on glass/ITO substrates. This PAL composition was chosen based on the $J_{sc}$, FF, and PCE vs thickness–composition landscapes established from lightly crosslinked P3HT networks[19]. Thirty-nm-thick Ca films, capped with 130-nm-thick Al films, were then thermally evaporated to give the electron collection contacts, which are expected to be ohmic[10].

Figure 2a shows the JV characteristics measured at an illumination intensity of 110 mW cm$^{-2}$, while Fig. 2b shows the

cell performance parameters plotted against $\phi$ of the HCL. All measurements were performed in a single batch to avoid calibration uncertainties. Four representative diodes are displayed for each HCL to show repeatability. Cells with $\phi = 5.17$ eV give FF = 0.63 ± 0.02, $J_{sc} = 9.4 \pm 0.3$ mA cm$^{-2}$, $V_{oc} = 0.595 \pm 0.01$ V, and PCE = 3.2 ± 0.2 %, similar to literature reports[10,25–28]. As $\phi$ decreases below 5.0 eV, the JV slope in the vicinity of $V_{oc}$ becomes noticeably gentler, indicating an increase in the open-circuit series resistance $\left( R_{s,oc} = \frac{dV}{dJ}\Big|_{V_{oc}} \right)$, even though $V_{oc}$ is unchanged. This shifts the maximum power point down-voltage, degrading both FF and PCE. As $\phi$ crosses ca. 4.73 eV, $V_{oc}$ downshifts with unity slope, as may be expected of a depinned hole contact[29]. Thus the apparent $\phi_{pin}$ is 4.73 eV. This has been confirmed by $V_{bi}$ measurements (vide infra). As soon as depinning occurs, the JV characteristic collapses to S shape[30–32], forcing PCE to zero within a narrow $\phi$ range of 0.1 eV, even though the photocurrent at reverse bias voltage of −1.0 V remains unchanged. Particularly relevant to cell optimization, FF increases with $\phi$ beyond $\phi_{pin}$ until 5.02 eV, at which it levels off, while $V_{oc}$ rolls off gently at a rate of −60 mV eV$^{-1}$. Therefore, $\phi_{op}$ exceeds $\phi_{pin}$ here by ca. 0.3 eV, and the electrode work function has to be brought to $\phi_{op}$ for best performance. The FF improvement from $\phi_{pin}$ to $\phi_{op}$ is primarily due to $R_{s,oc}$ reduction. While FF remains optimal beyond $\phi_{op}$, cell performance eventually degrades due to the $V_{oc}$ roll-off.

**Confirmation of $\phi_{pin}$ transition**. To confirm the nature of the $\phi_{pin}$ transition, we show here that $V_{bi}$ indeed becomes pinned at $\phi_{pin}$. We obtained $V_{bi}$ from $V_{oc}$ measurements at 1 sun, and below 150 K where shutdown of carrier injection leaves photocurrent to freely track the direction of the internal field, which reverses at the flatband condition[10,33]. Figure 3 shows the $V_{oc}$ ($\phi$) plots for selected temperatures. $V_{bi}$ exhibits a temperature dependence of the form: $V_{bi} = V_o - \beta T$, where $V_o$ is the zero-K term and $\beta$ is of the order of a few $k/e$[10]. $\beta$ receives contributions from the temperature dependences of bandgap and also electrostatic band

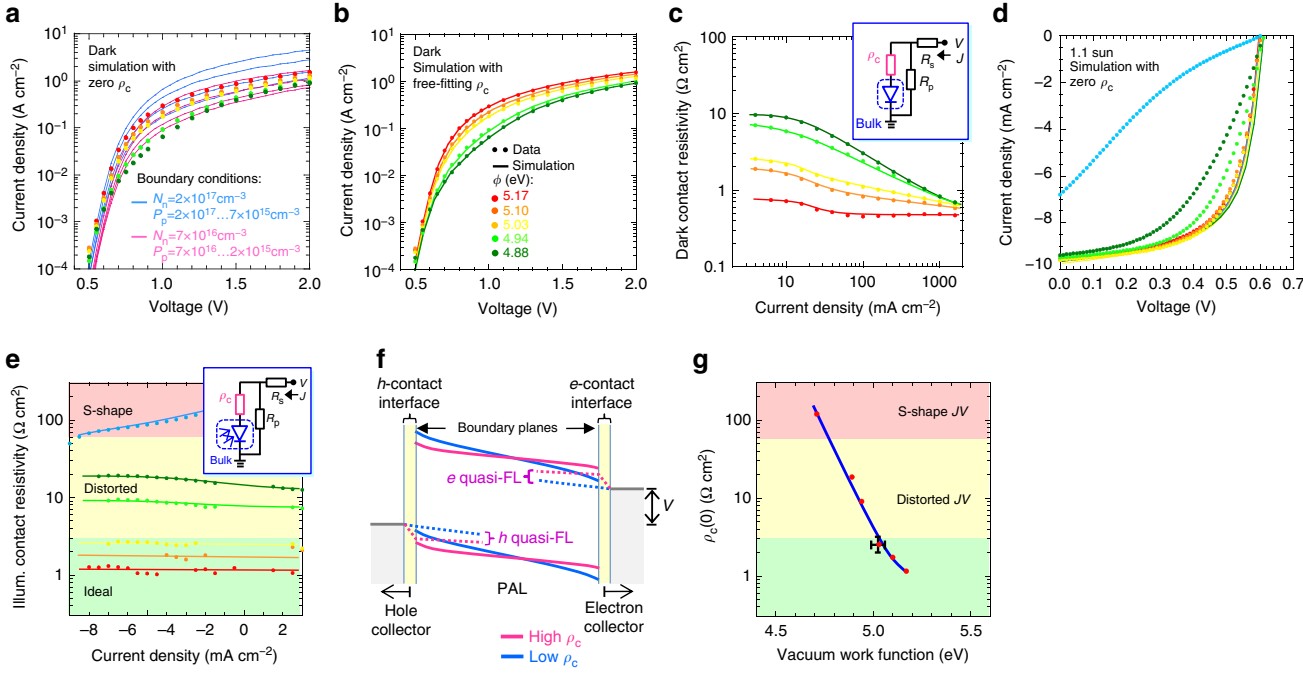

**Fig. 4** Simulation and modeling of contact resistivity. **a** Experimental and drift–diffusion-simulated dark $JV$ characteristics without contact resistance. Symbols are data; lines are simulations for the specified electron ($N_n$) and hole ($P_p$) densities at the electron- and hole-contact boundaries, respectively. Data color coding is given in **b**. Adjacent lines in each family (blue, magenta) differ by a factor of 3.16. **b** Same as **a** but with free-fitting contact resistivity function $\rho_c(J)$. Simulation parameters are given in Supplementary Table 1. **c** Dark $\rho_c(J)$ characteristics. Inset shows equivalent-circuit model. Symbols are data; lines are guide to the eye. Data color coding is same as **b**. **d** Experimental and drift–diffusion–generation-simulated illuminated $JV$ characteristics without contact resistance. Symbols are data; lines are simulations for the self-consistent $\{N_n, P_p\}$ found in **b**. Data color coding is same as **b**, with additional sky blue = 4.71 eV. **e** Illuminated $\rho_c(J)$ characteristics. Inset shows equivalent-circuit model. Data color codingis same as **d**. **f** Schematic energy-level diagram of the cell under illumination with external load, showing boundary planes and (expanded) contact interfaces, and illustrating the electron and hole quasi-Fermi levels in the near contact regions for high and low $\rho_c$. The drop in FL across the contact interface is due to contact resistance. **g** Zero-current total contact resistivity plotted against vacuum work function of the hole collection layer. Error bar gives typical standard error

bending due to the accumulated carrier tail[34,35]. This band bending has been confirmed by UPS on polymer organic semiconductors[36], including monolayer to multilayer stacks of P3HT on PEDT:PSSH[37]. Lowering temperature contracts this carrier tail, increasing $V_{bi}$. Thus the $V_{oc}$ ($\phi$) plots below 70 K collapse to a "universal" characteristic, indicating that $V_o$ has been approached within experimental resolution. This characteristic exhibits a unity slope (i.e., $\frac{dV_o}{d\phi} = 1$) that turns over to zero slope ($\frac{dV_o}{d\phi} = 0$) at a well-defined threshold, a classic signature of the FL pinning transition[15–17] (Supplementary Note 1). The threshold gives $\phi_{pin}$ for FL pinning to P3HT:PCBM to be 4.80 eV for temperatures <150 K. In contrast, FL pinning to edge-on, π-stacked P3HT lamellae occurs at 4.40 eV[38], presumably due to order and orientation effects. Linear extrapolation to room temperature (295 K) gives $V_{bi} = 0.66$ ($\pm 0.02$) V. This intersects with the temperature-independent (unity slope) segment of the plot to give $\phi_{pin} = 4.75$ ($\pm 0.02$) eV at room temperature. This confirms that the kinks in both $V_{oc}(\phi)$ and $J_{sc}(\phi)$ indeed occur at the FL pinning transition for room temperature.

**Extracting contact resistivity**. To understand the dependence of $R_{s,oc}$ on $\phi$, we first develop a general methodology to extract contact resistivity characteristic $\rho_c(J)$, and then show that this is a property of the contact, applicable to both dark and illuminated conditions. The methodology is based on drift–diffusion simulation of $JV$ characteristics with bimolecular bulk recombination in the PAL between its two surface boundaries next to the contacts[20,39], with optical transfer-matrix simulation of light

absorption within the PAL when illuminated[9,10]. We imposed globally self-consistent boundary conditions, and assigned the residual "excess" resistivity to the total contact resistivity of the semiconductor/electrode interfaces. For dark $JV$ characteristics, we computed the contact resistivity as: $\rho_c(J) = \frac{V(J) - V_{int}(J)}{J}$, where $V$ is the applied bias, and $V_{int}$ is the internal voltage across the PAL given by drift–diffusion simulation. Thus this approach is advantageously model-free[40–42]; it does not presume the diode equation nor any particular charge injection (or extraction) model. The self-consistent boundary conditions were found by systematic search for global best fit to the entire set of $JV$ characteristics (all $\phi$) in the diffusion-current regime, where $\rho_c$ can be neglected compared to bulk resistivity, subjected to the constraints that $N_n$ is constant, but $P_p \sim (\phi - \phi_{pin})$, where $N_n$ is the electron density at the boundary to the electron contact and $P_p$ is the hole density at the boundary to the hole contact[43]. All other input parameters are known from previous work and assumed constant[10,25] (see Supplementary Table 1).

We first point out that the dark $JV$ characteristics outside of the diffusion regime cannot be altogether fitted to any self-consistent $\{N_n, P_p\}$ set with zero $\rho_c(J)$ (Fig. 4a). The discrepancy is particularly severe for the cells with sub-optimal HCLs. This disparity can obviously be eliminated by including a free-fitting $\rho_c(J)$ function to represent excess resistivity (Fig. 4b). The requisite dark $\rho_c(J)$ characteristics are shown in Fig. 4c. These generally level off at low $J$ to a high limit that increases strongly with $\phi_{op} - \phi$, e.g., 0.7 Ω cm² at $\phi = 5.15$ eV increasing to 10 Ω cm² at $\phi = 4.9$ eV; they also level off at high $J$ to a low limit of ca. 0.5

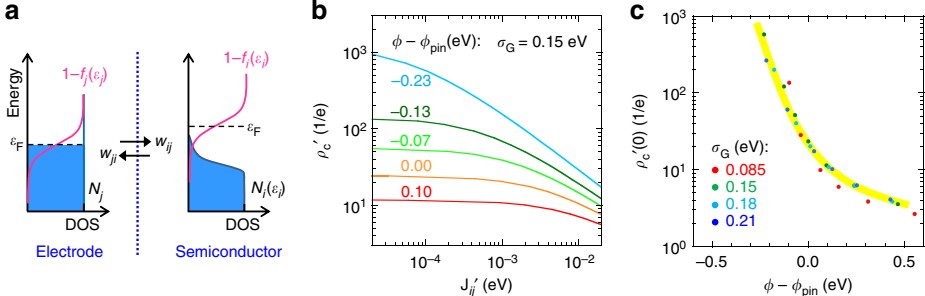

**Fig. 5** Current detailed balance analysis. **a** Schematic energy band model of hole contact under forward bias. **b** Plot of theoretical reduced contact resistivity $\rho_c'$ (in units of $1/e$) against reduced current density $J_{ij}'$, for different $\phi - \phi_{pin}$ values, and Gaussian width $\sigma_G = 0.15$ eV. **c** Plot of $\rho_c'(0)$ against $\phi - \phi_{pin}$. Symbols are computed results; yellow line is guide to the eye for $0.15 < \sigma_G < 0.2$ eV

$\Omega$ cm$^2$. Probe and lateral electrode resistances together contribute $<0.2\,\Omega$ cm$^2$, with the balance likely dominated by $\rho_c$ of the electron contact. Since this contact is unchanged, the strong $\phi$ dependence of $\rho_c(J)$ can directly be attributed to the hole contact. For $\phi \gtrsim 5.15$ eV, $\rho_c(J)$ shows little dependence on $J$, a non-dispersive behavior found also for notionally ohmic contacts in organic field-effect transistors[44,45]. Near $\phi_{pin}$ however, $\rho_c(J)$ exhibits an apparent inverse power law dependence, with exponent of $-0.67$, between the two limits.

Similarly, the illuminated $JV$ characteristics in the power-generation quadrant, particularly for cells with sub-optimal HCLs, also cannot be fitted to any self-consistent $\{N_n, P_p\}$ set with zero $\rho_c(J)$ (Fig. 4d). In this case, $V_{int}(J)$ is only slightly modified by the expected variation in $P_p$. So the marked dependence of $R_{s,oc}$ on $\phi$ provides clear evidence for contact resistance. For illuminated $JV$ characteristics, we computed this contact resistivity as: $\rho_c(J) = \frac{V_{int}(J) - V(J)}{J}$, where the order of terms is swapped due to the reverse direction of flow of the photocurrent. The illuminated $\rho_c(J)$ characteristics thus obtained are shown in Fig. 4e. The close match between the dark and illuminated $\rho_c$ at zero current $\rho_c(0)$, together with their strong $\phi$ and $J$ dependences, confirms that the excess resistivity is due to the contact resistivity at the semiconductor/electrode interface. This is not negligible until well beyond the threshold of FL pinning.

The FL drop at the contact caused by this $\rho_c(J)$ creates a barrier to charge extraction by lowering the internal field (Fig. 4f), promoting non-geminate recombination in the bulk of the semiconductor and near-surface region of the contact. Figure 4g plots $\rho_c(0)$ against $\phi$. It reveals an inverse exponential-like dependence: $\rho_c(0) \sim \exp(-(\phi - \phi_{pin}))$, decreasing from 100 $\Omega$ cm$^2$ at $\phi = 4.7$ eV to 1 $\Omega$ cm$^2$ at $\phi = 5.2$ eV, with $\left( \frac{d \log(\rho_c / \Omega cm^2)}{d\phi} \right)$ slope of ca. 5 eV$^{-1}$ in the vicinity of $\phi_{pin}$.

**Current detailed balance analysis.** To clarify the mechanistic origin of $\rho_c(J)$, we first formulate a contact resistance theory in the current detailed balance formalism and then show that this leads to an intrinsic charge-transfer resistance that quantitatively agrees with $\rho_c(J)$. Therefore $\rho_c(J)$ is a fundamental feature of hopping charge injection/collection. The current detailed balance model of the contact is shown in Fig. 5a. Contact resistivity is modeled by site-to-site hopping between the two frontier monolayers (denoted $i$ and $j$) across the contact. The net hole current density $J_{ij}$ from $j$ to $i$ under the influence of a small local bias $\partial V$, i.e., applied potential difference across the two monolayers of the contact (positive for forward bias), is given by detailed balance to

be:

$$J_{ij} = e \int_{i,j} \alpha \left[ w_{ij} N_i(\varepsilon_i) f_i(\varepsilon_i) N_j(\varepsilon_j)(1 - f_j(\varepsilon_j)) \right.$$
$$\left. - w_{ji} N_j(\varepsilon_j) f_j(\varepsilon_j) N_i(\varepsilon_i)(1 - f_i(\varepsilon_i)) \right] \delta(\varepsilon_i - \varepsilon_j + e\partial V) d\varepsilon$$

where $\alpha$ is the effective charge-hopping cross-section area; $w_{ij} = w_{ji}$ is the transfer integral given by Fermi golden rule: $w_{ij} = \frac{4\pi^2}{h} V^2 \chi^2$, where $V^2$ is the electronic Hamiltonian and $\chi^2$ is the Franck–Condon overlap; $N_j(\varepsilon_j)$ and $N_i(\varepsilon_i)$ are the density-of-state (DOS) per unit area per unit energy interval, generally different for $j$ and $i$; $\varepsilon_j$ and $\varepsilon_i$ are the respective (electron) energies; $f_j(\varepsilon_j)$ and $f_i(\varepsilon_i)$ are the usual Fermi–Dirac functions; and $\delta(\dots)$ is the Kronecker delta that enforces energy conservation during the hop. The corollary gives the electron current, if required. Assuming both $w_{ij}$ and $\alpha$ are independent of $\varepsilon_j$ and $\varepsilon_i$ over the small energy range relevant to injection or collection (i.e., $\pm 2kT$), we can write

$$J_{ij} = e\alpha w_{ij} N_{o,i} N_{o,j} \int_{i,j} \left[ \xi_i(\varepsilon - e\partial V) f_i(\varepsilon - e\partial V) \xi_j(\varepsilon) \left( 1 - f_j(\varepsilon) \right) \right.$$
$$\left. - \xi_j(\varepsilon) f_j(\varepsilon) \xi_i(\varepsilon - e\partial V)(1 - f_i(\varepsilon - e\partial V)) \right] d\varepsilon$$
$$= e\alpha w_{ij} N_{o,i} N_{o,j} J_{ij}',$$

where the product $N_{o,i} N_{o,j}$ is the joint DOS per unit area of $i$ and per unit area of $j$, per unit energy interval, such that $\xi(\varepsilon) = N(\varepsilon)/N_o$ and $J_{ij}'$ is the reduced current density (in units of energy). The zero-current contact resistivity is thus obtained from: $\rho_c(0) = \lim_{J_{ij} \to 0} \frac{\partial V}{J_{ij}}$. It is more instructive, however, to display the reduced contact resistivity: $\rho_c'(0) = \lim_{J_{ij}' \to 0} \frac{\partial V}{J_{ij}'}$, which is $\rho_c(0)$ normalized by $e\alpha w_{ij} N_{o,i} N_{o,j}$.

**Semiconductor with Heaviside DOS.** We first treat a simplistic contact between a heavily doped electrode $j$, modeled by constant $N_j(\varepsilon_j)$ over the relevant energy range in the vicinity of $\varepsilon_F$, and a semiconductor $i$, modeled by $N_i(\varepsilon_i)$ that is a Heaviside step function with transition edge at $\varepsilon_b$. This contact presents a thermodynamic barrier for hole injection given by: $E_b = \varepsilon_F - \varepsilon_b$. For $E_b > 2kT$ at zero $\partial V$, one can readily show that: $\rho_c(0) = (e^2 \alpha w_{ij} N_{o,i} N_{o,j})^{-1} \exp\left(\frac{E_b}{kT}\right)$. This equation reveals two key features. First, both injection and collection are subjected to the same $\rho_c(0)$; second, $\rho_c(0)$ is finite. The reason for the first feature is that carrier collection also requires FL offset as carrier injection. This leads to the following consequence: a good injection contact is also a good collection contact, and vice versa. An energy barrier that suppresses the electrode-to-semiconductor injection current

also suppresses the semiconductor-to-electrode collection current in the same way because of depletion of carriers in the semiconductor DOS.

**Semiconductor with Gaussian DOS**. We then treat a more realistic contact of the same electrode but with a disordered semiconductor characterized by a hemi-Gaussian tail with disorder width $\sigma_G$ in its frontier DOS. This requires numerical integration of the detailed balance equation. The results reveal that the two features above are preserved. Incidentally, this justifies the assumption of identical source and drain contact resistivities in organic field-effect transistors with symmetrical contacts[44,45]. To obtain the theoretical dependence of $\rho_c$ on $\phi$ and $\sigma_G$, we stepped FL through the DOS tail, evaluating electrode $\phi$ at each step and taking into account the carrier density accumulated at the contact (Supplementary Note 2). We found that, for $\sigma_G$ over the reasonable range of 0.15–0.2 eV, the computed form of $\rho_c{}'(J')$ matches experiment well without any fitting parameters. In particular, for $\phi \gtrsim \phi_{pin}$, $\rho_c{}'(J')$ is substantially flat; for $\phi$ just smaller than $\phi_{pin}$, $\rho_c{}'(J')$ levels off at low $J'$ but gives power law dependence with exponent –0.67 at higher $J'$ (Fig. 5b). The computed $\rho_c{}'(0)(\phi)$ characteristic also exhibits a form similar to experiment, with $\left(\frac{d\log(\rho_c{}')}{d\phi}\right)$ slope of 5 eV$^{-1}$ near $\phi_{pin}$ (Fig. 5c). For reasonable parameters: $N_{o,i}N_{o,j} \approx 1 \times 10^{25}$ cm$^{-4}$ eV$^{-1}$, a $\approx 30$ Å$^2$, $w_{ij} \approx 1 \times 10^{10}$ s$^{-1}$, the $J_{ij}' = 1$ eV level corresponds to 100 A cm$^{-2}$, and $\rho_c{}' = 1\, e^{-1}$ level corresponds to 0.03 Ω cm$^2$. This agrees with experiment to within half an order of magnitude. Therefore, the measured $\rho_c(J,\phi)$ characteristics are indeed manifestations of the hopping injection/extraction process at a disordered semiconductor contact.

Nevertheless, the steep dependence of $\rho_c(0)$ on $\phi$ extends further beyond $\phi_{pin}$ than predicted by theory. This suggests a possible $\phi$ dependence in $w_{ij}$. Subgap electroabsorption spectroscopy found spectral changes that suggest polaron interactions within the accumulation layer at carrier densities above a few $10^{11}$ hole cm$^{-2}$ [33]. The results here suggest that $w_{ij}$ also becomes larger, which may arise, for example, if the doped polymer segments exhibit a higher $w_{ij}$ than the undoped segments.

**Ohmic transition**. The inverse exponential-like dependence of $\rho_c$ on $\phi$ produces a fairly sharp transition that corresponds to emergence of ohmic contact in the device when $\phi$ is tuned over a narrow range across $\phi_{op}$. This can be characterized as the ohmic transition, akin to the glass transition that follows from the inverse exponential dependence of viscosity on temperature. Signatures of such behavior may be discerned in data on other systems, even though the limited resolution and range of earlier work have precluded analysis[46–49]. Whether a contact is ohmic or not, however, is determined by whether its resistivity is small compared to that of the bulk of the device. Therefore the precise location of the ohmic transition depends on bulk resistivity. One may thus specify a total $\rho_c$ criterion for the ohmic transition at which device behavior is no longer dominated by contacts.

For solar cells, one may specify that $\rho_c$ needs to be smaller than about half of the bulk resistivity at $V_{oc}$, i.e., $\rho_c(V_{oc}) < \frac{1}{2}\frac{dV_{int}}{dJ}\Big|_{V_{oc}}$.

Drift–diffusion modeling indicates for typical P3HT:PCBM cells under 1 sun, the bulk resistivity is ca. 5.5 Ω cm$^2$ in the vicinity of $V_{oc}$. Thus assuming the electron-contact $\rho_c$ is negligible, the hole-contact $\rho_c$ needs to be smaller than ca. 3 Ω cm$^2$. Two phenomenological regimes can then be distinguished—a bulk-limited regime for $\phi \gtrsim \phi_{op}$ where $\rho_c \lesssim 3$ Ω cm$^2$; and a contact-limited regime for $\phi < \phi_{op}$, where $J(V)$ is at first lightly degraded ($\phi_{pin} \lesssim \phi \lesssim \phi_{op}$; $3 \lesssim \rho_c \lesssim 50$ Ω cm$^2$) and then heavily degraded ($\phi \lesssim \phi_{pin}$; $\rho_c \gtrsim 50$ Ω cm$^2$). For injection diodes on the other hand, one may specify that $\rho_c$ needs to be smaller than about half of the bulk resistivity at the desired $J$, i.e., $\rho_c(J) < \frac{1}{2}\frac{dV_{int}}{dJ}\Big|_J$. For P3HT:PCBM diodes with $N_n$, $P_p \gtrsim 2 \times 10^{17}$ cm$^{-3}$, the computed bulk resistivity is 7.5 Ω cm$^2$ at $J = 100$ mA cm$^{-2}$. Therefore, assuming again the electron-contact $\rho_c$ is negligible, the hole-contact $\rho_c$ needs to be smaller than ca. 4 Ω cm$^2$ at this current density.

**Conclusion**. In summary, we have successfully developed a model PEDT:PSS(Cs$_x$H$_{1-x}$) electrode system whose inherent work function can be continuously tuned over a wide range without changing morphology, surface dipole or doping level. This enables us to show that the work function of the hole collection electrode that is needed to maximize the FF of organic solar cells is larger than that required to maximize their open-circuit voltage by few tenths of an eV. This arises as a consequence of the charge-transfer contact resistance at the semiconductor/electrode interface continuing to decline strongly with work function even in the Fermi-level pinned regime, due to the increasing accumulation of carrier density at the semiconductor side of the contact. Hence the ohmic transition occurs beyond the Fermi-level pinning transition. Although particularly important for solar cells, which are highly sensitive to the resulting series resistance, the clarification here of the ohmic transition closes a key gap in our understanding of disordered semiconductor contacts, pointing to a simple rule for the design of contacts relevant also to injection diodes and field-effect transistors. The development of heavily doped contacts, whether imposed by charge-counterbalancing polyelectrolyte monolayers[45], or the use of self-compensated, heavily doped polymer injection layers[50], is thus key to realizing ohmic contacts at will.

## Methods

**Preparation of PEDT:PSSCs$_x$H$_{1-x}$ hole collection electrode films**. PEDT:PSSCs$_x$H$_{1-x}$ solutions were prepared by mixing PEDT:PSSH and PEDT:PSSCs solutions in the desired stoichiometric ratio, filtering through a 0.45-μm filter, and spin-casting to give the desired films. The PEDT:PSSCs solution was prepared from commercial PEDT:PSSH solution (Heraeus P VP AI4083, 1:6 w/w) by ion-exchange dialysis[51]. The PEDT:PSSH solution was first rigorously purified of ions by dialysis[52]. Rigorous ion removal is important to obtain reproducible results. Twenty-nm-thick PEDT:PSSCs$_x$H$_{1-x}$ films were spin-cast on Au-coated Si substrates, annealed at 140 °C (hotplate, 10 min) in a N$_2$ glovebox ($pO_2$, H$_2$O < 1 ppm) to remove water, and transferred in N$_2$ to the UPS chamber equipped with an Omicron EA 125 energy analyzer and five channeltrons for $\phi$ measurements.

**Diode fabrication and measurements**. Standard P3HT:PCBM diodes were fabricated by spinning 85-nm thick P3HT:PCBM films (1.0:0.8 w/w) over PEDT:PSSCs$_x$H$_{1-x}$ films. The films were annealed at 140 °C (hotplate, 10 min) in a N$_2$ glovebox before and after PAL deposition to remove water and solvent, respectively. Thirty-nm-thick Ca capped with 130-nm-thick Al were then evaporated to define 4.2-mm$^2$ electron-collection electrodes.

**Data availability**. The data supporting the findings of this study are available within the article and its Supplementary Information files.

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

## Acknowledgements

The authors acknowledge Ministry of Education, Singapore for financial support (R-144-000-324-112). This research is partially supported by National Research Foundation, Prime Minister's Office, Singapore under its competitive research programme (CRP Award No. NRF-CRP 11-2012-03: R-144-000-339-281). The Solar Energy Research Institute of Singapore (SERIS) is sponsored by the National University of Singapore (NUS) and the National Research Foundation (NRF) of Singapore through the Singapore Economic Development Board (EDB).

## Author contributions

J.K.T. collected and interpreted the data. J.K.T. and P.K.H.H. performed the drift–diffusion simulations, C.Z. and P.K.H.H. performed the current detailed balance calculations. R.Q.P and P.K.H.H. directed the work, formulated the theory, and wrote the report.

**Additional information**

**Competing interests:** The authors declare no competing interests.

