## [Peer Review File · Nature Communications]

Reviewers' comments:

Reviewer #1 (Remarks to the Author):

In this paper, Jun-Kai Tan, et al studied on the energy-level alignment at a model contact for the organic solar cells with P3HT: PCBM via adjustable work function system, PEDT: PSSCsxH1-x composition conducting polymers. Contact resistivity diminishes exponentially with further improvement in work function of electrode interface at the pinning threshold, which has important implications for maximizing PCE in organic solar cells. The paper demonstrates satisfactory effects in the devices and provide a solid physical explanation for how the contact engineering works. The contact model is well conducted, of certain relevance to the field. However, by very nature of interface semiconductors (metal oxides, conducting polymer, small molecules and so on.), the contact model within this paper are limited, though this is available to P3HT:PCBM solar cells with PEDT: PSSCsxH1-x composition conducting polymers. For example, metal oxides with different levels of doping also could tune the surface work function, meanwhile increase contact resistivity of metal oxides. However, performance of solar cells has not improved. It seems to me that the authors did not try to distinguish the nature of different interface semiconductors to understand ohmic contacts for organic electronics devices.

Moreover, authors simulate the J-V for dark and light injection with non-zero and zero contact resistivity by drift-diffusion-generation modeling to agree with the effect of contact resistivity for charge injection. However, physical process of interface is complicated with surface recombination kinetics. Authors did not carefully understand the underline of physical process including charge injection, extraction and transportation.

Authors claimed "Cs+ substitution downshifts work function by reducing Coulomb repulsion between hole carriers and spectator cations, without altering morphology nor doping level of the PEDT chains.

" Hence, "photocurrent collection are determined by contacts rather than interior dopant-diffused junctions". It seems to me that the authors did not more explanation or experimental results. Additionally, through H+ and Cs+ ionic-exchange in the PEDT: PSSCsxH1-x composition conducting polymer, maybe there were electron doping because of the stronger electronegativity of Cs. After a couple of issues need to be addressed, this contact model for the organic solar cells could be able to generate critical impact.

Reviewer #2 (Remarks to the Author):

In the manuscript the effect of a contact barrier on the performance of an organic solar cell is investigated. What is special in this study is that the electrode work function is continuously tuned over a 1 eV range. It appears that above a certain barrier height (~ 0.3 eV) there is a threshold where a contact resistance sets in, which grows exponentially with increasing barrier height. This is an important finding to understand the demands on contact barriers in organic solar cells. I have two minor comments/questions:

- for the dark current simulations the hole/electron density at the contact is varied at the contacts. In drift-diffusion simulations typically an effective density of states is used that relates the carrier density to the barrier height. Organic semiconductors are disordered, characterized by a Gaussian density of states. This provides a different coupling between carrier density and barrier height. Assuming a width of 0.1 eV for every density the Fermi-level in the DOS can be calculated and its distance to the centre could be used as barrier height. Does the incorporation of a Gaussian DOS improve the agreements with the simulations for the dark and photocurrent?
- the transition from ohmic to non-ohmic can also be observed in the dependence of Voc on light

intensity, see Solak et al., Appl. Phys. Lett. 109, 053302 (2016); doi: 10.1063/1.4960151. It would be new if the authors could show if the transition in slope from kT/q to $kT/2q$ is gradual or makes a jump at the transition point.

Referee #1

We thank this Referee for his/ her critical comments to improve the manuscript. We have revised accordingly, completing the theoretical formulation of the charge-injection (“contact”) resistance that yields new physical insight into the process. We would thus like to respond to the comments as follows.

1. However by very nature of interface semiconductors (metal oxides, conducting polymer, small molecules and so on.), the contact model within this paper are limited, though this is available to P3HT:PCBM solar cells with PEDT: PSSCs_xH_{1-x} composition conducting polymers. For example, metal oxides with different levels of doping also could tune the surface work function, meanwhile increase contact resistivity of metal oxides. However, performance of solar cells has not improved. It seems to me that the authors did not try to distinguish the nature of different interface semiconductors to understand ohmic contacts for organic electronics devices.

Response. We developed the PEDT:PSS(CS_xH_{1-x}) system precisely to achieve change of work function without change of doping level. Changing doping level can change the work function, but this causes changes not only in carrier density, but also surface band bending (i.e., depletion width) and electrical conductivity (e.g. Zhou PCCP 2012 14 12014), which complicates analysis. Here we develop a “clean” PEDT: PSSCs_xH_{1-x} system where work function can be manipulated independently of doping level and morphology. The new data included in Supplementary Figs 1a and 1b show no shift in valence band features of PEDT, PSS and Cs⁺ vs vacuum level as x increases, confirming that there is no differential surface dipole contribution to the vacuum work function shift. The valence density-of-states band edge shifts rigidly with Fermi level, showing that the work function change is not associated with a change in doping level or PEDT:PSS morphology (Fig. 1b). As a consequence, this system provides for the first time a robust “model” contact where work function can be cleanly manipulated without the usual complications from changing doping level, allowing us to investigate work function effects. In the revised manuscript, we have now also completed formulation of the current detailed balance theory, outlined in our original manuscript, to show that the measured results are in quantitative agreement with theory, providing firm theoretical basis for our earlier assertion that there exists an extraction contact resistance that arises fundamentally from the same consideration as the injection resistance.

2. Moreover, authors simulate the J-V for dark and light injection with non-zero and zero contact resistivity by drift–diffusion–generation modeling to agree with the effect of contact resistivity for charge injection. However, physical process of interface is complicated with surface recombination kinetics. Authors did not carefully understand the underline of physical process including charge injection, extraction and transportation.

Response. We have now completed the formulation of the current detailed balance theory on interfacial currents that now reveals a symmetry in the contact resistance charge injection and extraction, providing the final theoretical justification to our conclusions originally based on experimental analysis. As the Referee knows very well, work function matching to transport level is well known in the field. However, this picture misses the important distinction of Fermi-level pinning threshold where V_{bi} levels off, and the nonohmic-to-ohmic contact transition discovered here, which is responsible for the continued improvement in fill factor until well past the Fermi-level pinning threshold. Both of these are new vital physical insights that have been missing in the field. Drift–diffusion–generation modelling takes care of bulk photogeneration, transport, and recombination, including surface recombination, self-consistently. Different from silicon solar cells, the “surface” recombination in organic solar cells is in fact dominated by recombination with the majority carrier density accumulated at the contacts, which is taken

into account directly in drift–diffusion–generation modelling (Adv En Mater 4 (2014) 1200972). Therefore all key factors have already been included appropriately.

3. Authors claimed “Cs⁺ substitution downshifts work function by reducing Coulomb repulsion between hole carriers and spectator cations, without altering morphology nor doping level of the PEDT chains. Hence, “photocarrier collection are determined by contacts rather than interior dopant-diffused junctions”. It seems to me that the authors did not more explanation or experimental results. Additionally, through H⁺ and Cs⁺ ionic-exchange in the PEDT: PSSCs_xH_{1-x} composition conducting polymer, maybe there were electron doping because of the stronger electronegativity of Cs.

Response. The Madelung potential effect has been established by the PRL2009 paper which we have referenced. That paper provided the full set of evidence on how changing from H to Cs does not change doping level nor morphology of PEDT chains. Furthermore there is no electron doping of the PEDT. This is clear from the data in the PRL2009 paper, which showed unchanged hole polaron electronic transition and infrared vibration (IRAV) bands. We have now included in Supplementary Figure 1 UPS data that shows no change in surface dipole moment across the PEDT:PSSCs_xH_{1-x}. Therefore this system behaves identically as the previous study, but has the further advantage that Cs⁺ is inert.

This Cs⁺ ion (not Cs metal!) is directly introduced into the system by ion exchange, so it cannot transfer any electron to the PEDT. Figure 1b shows there is no change in the electron density-of-states distribution in the vicinity of the Fermi energy, hence there is no electron doping.

The statement that “photocarrier collection are determined by contacts rather than interior dopant-diffused junctions” refers to the fact that the built-in potential of the devices are determined by the contacts, and not an internal p-i-n structure as in Si.

Referee #2

We thank this Referee for his/ her excellent insights and critique of our manuscript. Prompted by the Referee, we have now completed the theoretical formulation of the interfacial charge-injection resistance problem outlined in the original manuscript, yielding new current detailed balance insights into the question.

1. I have two minor comments/questions:

-for the dark current simulations the hole/electron density at the contact is varied at the contacts. In drift diffusion simulations typically an effective density of states is used that relates the carrier density to the barrier height. Organic semiconductors are disordered, characterized by a Gaussian density of states. This provides a different coupling between carrier density and barrier height. Assuming a width of 0.1 eV for every density the Fermi-level in the DOS can be calculated and its distance to the centre could be used as barrier height. Does the incorporation of a Gaussian DOS improve the agreements with the simulations for the dark and photocurrent?

Response: We thank this Reviewer for this incisive comment that prompted us to complete our formulation of the theoretical understanding of contact resistance at disordered interfaces through the current detailed balance theory. As a consequence, we incurred some time to do this. The drift-diffusion-generation treats an effective mobility for carriers at a single effective level, within the bulk (thermal equilibrium approximation). In order to treat an extended DOS for injection, we considered site-to-site hopping across the injection contact in the current detailed balance formalism. This takes care of coupling at different energies across the contact. We treated both a simplistic Heaviside DOS function and a more realistic Gaussian DOS (as suggested by Reviewer). We are delighted to note that the simulated characteristics agree with the experiment for reasonable Gaussian widths. The section on “Theory: Current detailed balance analysis” have now been completely updated. New presentation images are provide in Figs 4h-j.

2. -the transition from ohmic to non-ohmic can also be observed in the dependence of Voc on light intensity, see Solak et al., Appl. Phys. Lett. 109, 053302 (2016); doi: 10.1063/1.4960151. It would be new if the authors could show if the transition in slope from kT/q to $kT/2q$ is gradual or makes a jump at the transition point.

Response: We thank this reviewer again for directing our attention to this paper. We missed it earlier, and have now included in the manuscript. This is an excellent paper that captured a large part in the state-of-the-art thinking on the Voc of organic solar cells, emphasizing the importance of V_{bi} , which we believe also to provide key physical insight. Our work now provides a firm understanding that this non-ohmic to ohmic transition occurs in two steps, a Fermi level pinning transition which “pins” Voc through the V_{bi} (which the community sort of “knew” although it never had such detailed data), followed by the actual and more important non-ohmic to ohmic transition in contact resistance (which is unknown).

The paper described a change in the light dependence $d(Voc)/d(\ln L)$ slope from kT/e for both contacts “ohmic” (presumably in the Fermi-level pinning regime) to $\frac{1}{2} * kT/e$ for one contact “ohmic”. We still don’t have the full data set to check this. The device with both contacts “ohmic” indeed shows a slope close to theory, 70 mV per decade vs 60 mV per decade (theory), across the initial V_{bi} , which extends the original observations of the Solak APL(2016) paper, but

bends over at very high sun intensity (> 10 sun), all measurements kept brief to avoid device heating. This bend over might be expected owing to quasi-Fermi level creep up the band edge, which would result in breakdown of Eqn (1) in the APL(2016) paper. The accurately unity slope in the plot of V_{oc} vs work function before Fermi-level pinning and its temperature independence suggest that V_{bi} indeed limits the V_{oc} . Such issues however may be require more thoughts, and we exclude them from present discussions in the manuscript.

REVIEWERS' COMMENTS:

Reviewer #1 (Remarks to the Author):

In this revised manuscript, the authors had successfully developed a model PEDT:PSS(CsxH1-x) electrode system whose inherent work function can be continuously tuned over a wide range, provided an evidence of a non-ohmic-to-ohmic transition in the charge-carrier collection contact resistance which has very important influence on the parameters of organic solar cells. Meanwhile, simulation with drift-diffusion-generation modelling is providing the theoretical justification to experimental results. Additionally, the disorder of semiconductor with DOS is a main reason of contact resistance. This revised manuscript brings a new surface contact model in the organic solar cells to provide a path for the understanding and designed contacts and therefore a few of contribution may be in this simple model. Moreover, this work also led an expansion in their Nature work, the doped polymer semiconductors with ultrahigh and ultralow work functions.

If there is any drawback to this it is only that neither the new PEDT:PSS(CsxH1-x) electrode, nor the effective FL pinning threshold ϕ_{pin} , BUT it does show that this model is rough, and some fundamental processes are not elaborated, such as the intermolecular interactions, the photo-induced charge and energy-transfer processes and bimolecular recombination kinetics at the contact. If I could recommend any changes to the model it would be to expound more on charge transfer kinetics at contacts - so that the reader has a broader perspective on the significance of this work.

From my point of view, the work is well-done and provides a simple effective model to the understanding of contact resistance at disordered interfaces and thus it merits to be published. Just, I suggest some minor modifications before publication: authors refer to sub-surface regions in the text, however you didn't define its regions.